# Current Cut Points of Three Falls Risk Assessment Tools Are Inferior to Calculated Cut Points in Geriatric Evaluation and Management Units

Vivian Lee [1], Linda Appiah-Kubi [1], Sara Vogrin [2], Jesse Zanker [2,*] and Joanna Mitropoulos [1]

1    Department of Geriatric Medicine, Western Health, Melbourne, VIC 3011, Australia;
vivianclee48@gmail.com (V.L.); linda.appiah-kubi2@wh.org.au (L.A.-K.);
joanna.mitropoulos@wh.org.au (J.M.)

2    Australian Institute for Musculoskeletal Science, AIMSS, Melbourne, VIC 3011, Australia;
sara.vogrin@unimelb.edu.au

*    Correspondence: jesse.zanker@unimelb.edu.au

**Abstract:** Background: Falls risk assessment tools are used in hospital inpatient settings to identify patients at increased risk of falls to guide and target interventions for fall prevention. In 2022, Western Health, Melbourne, Australia, introduced a new falls risk assessment tool, the Western Health St. Thomas' Risk Assessment Tool (WH-STRATIFY), which adapted The Northern Hospital's risk tool (TNH-STRATIFY) by adding non-English speaking background and falls-risk medication domains to reflect patient demographics. WH-STRATIFY replaced Peninsula Health Risk Screening Tool (PH-FRAT) previously in use at Western Health. This study compared the predictive accuracy of the three falls risk assessment tools in an older inpatient high-risk population. Aims: To determine the predictive accuracy of three falls risk assessment tools (PH-FRAT, TNH-STRATIFY, and WH-STRATIFY) on admission to Geriatric Evaluation Management (GEM) units (subacute inpatient wards where the most frail and older patients rehabilitate under a multi-disciplinary team). Method: A retrospective observational study was conducted on four GEM units. Data was collected on 54 consecutive patients who fell during admission and 62 randomly sampled patients who did not fall between December 2020 and June 2021. Participants were scored against three falls risk assessment tools. The event rate Youden (Youden Index$^{ER}$) indices were calculated and compared using default and optimal cut points to determine which tool was most accurate for predicting falls. Results: Overall, all tools had low predictive accuracy for falls. Using default cut points to compare falls assessment tools, TNH-STRATIFY had the highest predictive accuracy (Youden Index$^{ER}$ = 0.20, 95% confidence interval CI = 0.07, 0.34). The PH-FRAT (Youden Index$^{ER}$ = 0.01 and 95% CI = −0.04, 0.05) and WH-STRATIFY (Youden Index$^{ER}$ = 0.00 and 95% CI = −0.04, 0.03) were statistically equivalent and not predictive of falls compared to TNH-STRATIFY. When calculated optimal cut points were applied, predictive accuracy improved for PH-FRAT (Cut point 17, Youden Index$^{ER}$ = 0.14 and 95% CI = 0.01, 0.29) and WH-STRATIFY (Cut point 7, Youden Index$^{ER}$ = 0.18 and 95% CI = 0.00, 0.35). Conclusions: TNH-STRATIFY had the highest predictive accuracy for falls. The predictive accuracy of WH-STRATIFY improved and was significant when the calculated optimal cut point was applied. The optimal cut points of falls risk assessment tools should be determined and validated in different clinical settings to optimise local predictive accuracy, enabling targeted fall risk mitigation strategies and resource allocation.

**Keywords:** falls; falls risk assessment tools; GEM

## 1. Introduction

    Inpatient falls are the most commonly reported incidents in many hospitals with higher risk in older adult sub-acute patients [1–5]. A *fall* is defined as "an event which results in a person coming to rest inadvertently on the ground or floor or other lower level,"

the definition widely accepted and used by the World Health Organisation [6–9]. One Australian study reported over 40% of patients have experienced at least one fall during their admission [10]. As falls lead to health complications for patients (both physical and psychological) [11], and greater utilisation of hospital resources [12], falls risk assessment tools have been used as part of a broader plan to reduce the risk of falls for patients in sub-acute wards [13]. The major risk factors for inpatient falls include delirium, cognitive impairment, previous falls, neurological disorders, and sensory impairments [14–17]. The main reason for an older adult's admission to Geriatric Evaluation and Management (GEM) units is to improve mobility and function prior to discharge [17]. GEM patients are typically frail with multiple co-morbidities and a high risk of falls [17].

The term *falls risk assessment tool* has been used to describe a class of diagnostic processes to manage falls risk [3]. These include *risk-factor checklists* that prompt healthcare workers to identify common modifiable fall risk factors to reduce harm through targeted plan development. *Numerical risk prediction tools* have cut points on a scale designed to predict the risk of future falls by calculating a score from a set of risk factors.

In 2022, a global multidisciplinary group presented consensus recommendations promoting the use of multifactorial falls risk assessments in preference to numerical falls risk screening [18]. This included recommendations that assessments, interventions, and strategies should consider local context and resources [18]. Numerical risk assessment tools, however, can form part of a multifactorial risk assessment, and due to the retrospective nature of our study, we sought to assess the accuracy of numerical falls risk screening tools.

Western Health previously used the Peninsula Health Falls Risk Screening Tool (PH-FRAT) (Appendix A) first developed in 1999 by the Peninsula Health Falls Prevention Service. The PH-FRAT is used by approximately 400 agencies worldwide [6] in a variety of settings including sub-acute care [6–8]. However, its predictive performance for falls has been found to be poor [6]. The St. Thomas Risk Assessment Tool in Falling elderly inpatients (STRATIFY) is the most widely studied falls risk assessment tool and has the best diagnostic validity [6]. The Northern Hospital Modified St Thomas's Risk Assessment Tool (TNH-STRATIFY) was modified from STRATIFY based on local data and included additional risk factors: age, impaired balance, drug and alcohol-related problems, and broadening of the agitation item to include confusion, intellectually challenged or impulsivity. These modifications resulted in statistically significant improvements to the predictive accuracy of TNH-STRATIFY compared to the original STRATIFY tool [19].

In 2021, the Western Health Falls Working Group modified the TNH-STRATIFY to form a new tool (WH-STRATIFY, Appendix A) by adding two additional risk factors: Non-English-Speaking Background (NESB) and medications affecting mobility (sedatives, antidepressants, anti-Parkinson's, diuretics, anti-hypertensives, hypnotics, and opioids), based on evidence of these medications increasing falls risk [6,7]. The NESB category was added to the WH-STRATIFY [6] based on the expectation that NESB leads to greater disadvantage in communication and education on falls risk management [6,7]. A difference between WH-STRATIFY and its predecessors is the inclusion of suggested management strategies directly linked to each identified fall risk factor. In addition to numerical scoring, WH-STRATIFY promotes interventions tailored to the patient's individual risk, aligned with current guidelines [18]. In 2022, WH-STRATIFY was launched at Western Health. We sought to undertake the first study to validate and assess the predictive accuracy of the WH-STRATIFY tool and assess whether adding local demographic risk factors such as non-English speaking background which is prevalent in Western Health, improves the tool's accuracy.

The falls risk assessment tools should be tested in clinical practice for validity and feasibility prior to use [6]. This includes comparing predictive accuracy to other falls risk assessment tools [6–9] and establishing the optimal cut points (i.e., the threshold at which a falls risk assessment tool predicts a fall [6,8,9,20]. In this study, a cut point is the minimum score required on a falls risk assessment tool to achieve the classification of someone predicted to have a fall during their GEM admission. Predictive accuracy varies with the

cut point for different populations suggesting the cut point should be validated in the setting where the tool is applied [6]. TNH-STRATIFY and PH-FRAT have been validated in other settings but their current cut-points could be optimised to improve predictive accuracy. We hypothesised that local validation will optimise the predictive accuracy of falls risk assessment tools.

*Aims*

This study of GEM unit inpatients compares the predictive accuracy for falls of the PH-FRAT, TNH-STRATIFY, and WH-STRATIFY risk assessment tools using default and calculated optimal cut points [17]. We chose to assess participants admitted to GEM because this patient population has an established high risk for falls.

## 2. Results of the Study

### 2.1. Demographics

A total of 116 participants, comprised of 54 fallers and 62 non-fallers with mean age 81.0 years (fallers) and 79.3 years (non-fallers), *p*-value = 0.284. Fallers had a significantly higher average length of stay than non-fallers (28.0 days compared to 13.7 days, *p*-value < 0.0001 (Table 1).

**Table 1.** Patient profiles.

| Profile | All | Fallers | Non-Fallers | *p*-Value |
|---|---|---|---|---|
| Number (%) | 116 (100%) | 54 (47%) | 62 (53%) | 0.46 |
| Mean age years (SD) | 80.10 (8.6) | 81.00 (8.40) | 79.30 (8.80) | 0.28 |
| Mean length of stay in days (SD) | 20.30 (16.9) | 28.00 (20.20) | 13.70 (9.40) | <0.01 |
| Male (%) | 50 (43%) | 28 (52%) | 22 (35%) | 0.08 |

### 2.2. Comparing Predictive Accuracy Using Default Cut Points

Figures 1–3 summarise the number of fallers and non-fallers for the three falls risk assessment tools using default cut points. Default cut points are the original scores for each tool that denote a patient with a high or low fall risk (e.g., PH-FRAT = 12; TNH-STRATIFY = 3; WH-STRATIFY = 3).

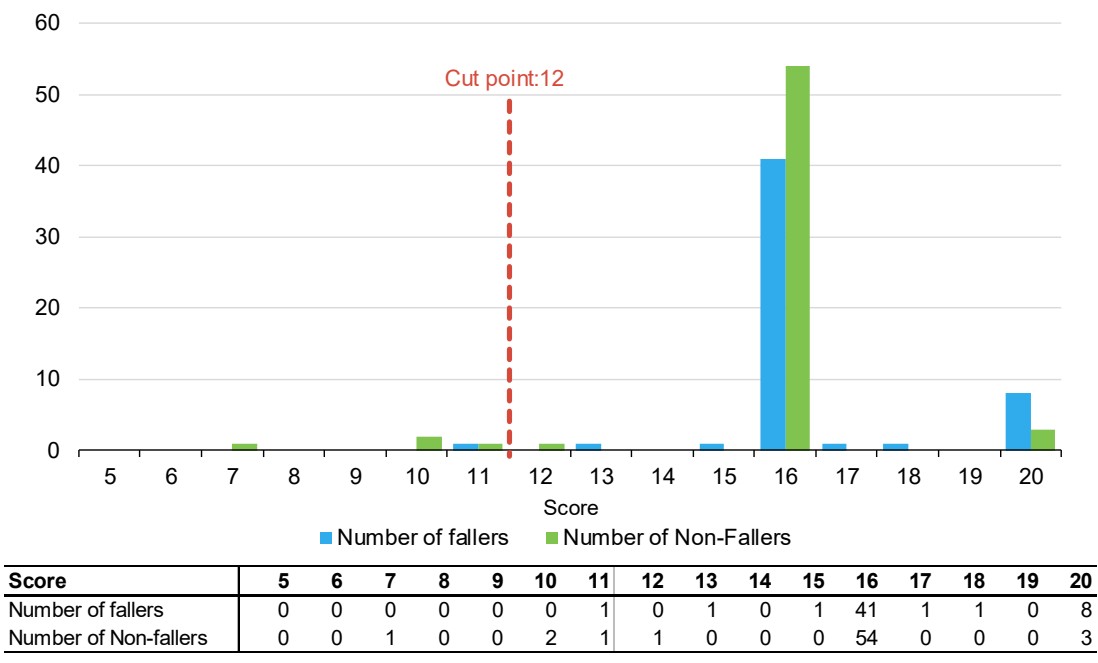

| Score | 5 | 6 | 7 | 8 | 9 | 10 | 11 | 12 | 13 | 14 | 15 | 16 | 17 | 18 | 19 | 20 |
|---|---|---|---|---|---|---|---|---|---|---|---|---|---|---|---|---|
| Number of fallers | 0 | 0 | 0 | 0 | 0 | 0 | 1 | 0 | 1 | 0 | 1 | 41 | 1 | 1 | 0 | 8 |
| Number of Non-fallers | 0 | 0 | 1 | 0 | 0 | 2 | 1 | 1 | 0 | 0 | 0 | 54 | 0 | 0 | 0 | 3 |

**Figure 1.** Number of fallers and non-fallers by PH-FRAT score.

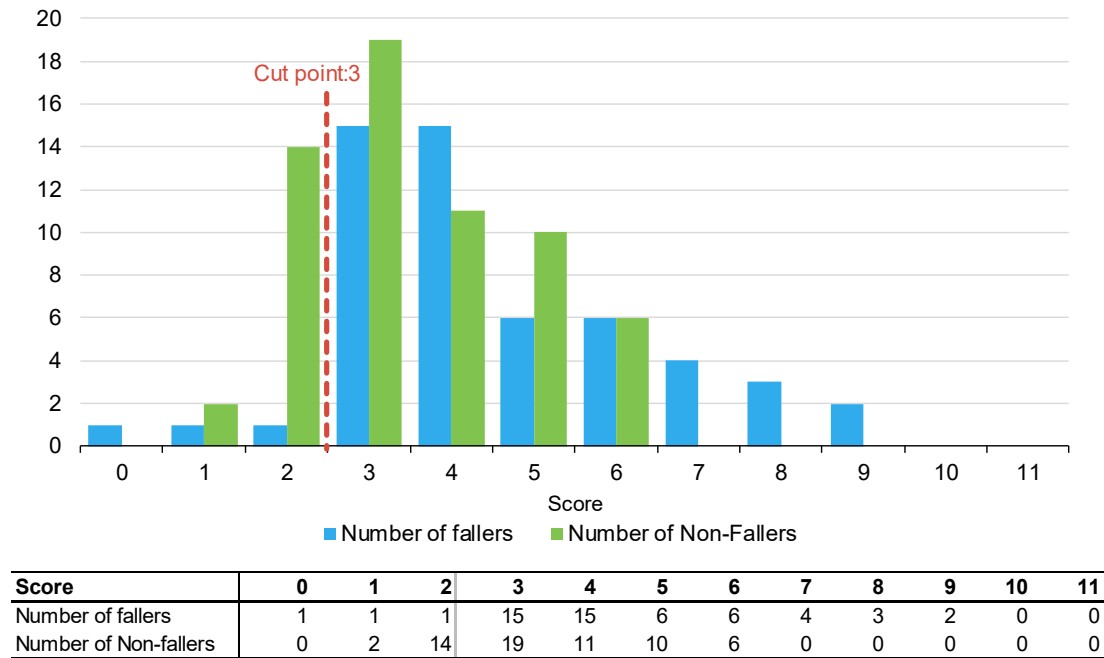

| Score | 0 | 1 | 2 | 3 | 4 | 5 | 6 | 7 | 8 | 9 | 10 | 11 |
|---|---|---|---|---|---|---|---|---|---|---|---|---|
| Number of fallers | 1 | 1 | 1 | 15 | 15 | 6 | 6 | 4 | 3 | 2 | 0 | 0 |
| Number of Non-fallers | 0 | 2 | 14 | 19 | 11 | 10 | 6 | 0 | 0 | 0 | 0 | 0 |

**Figure 2.** Number of fallers and non-fallers by TNH-STRATIFY score.

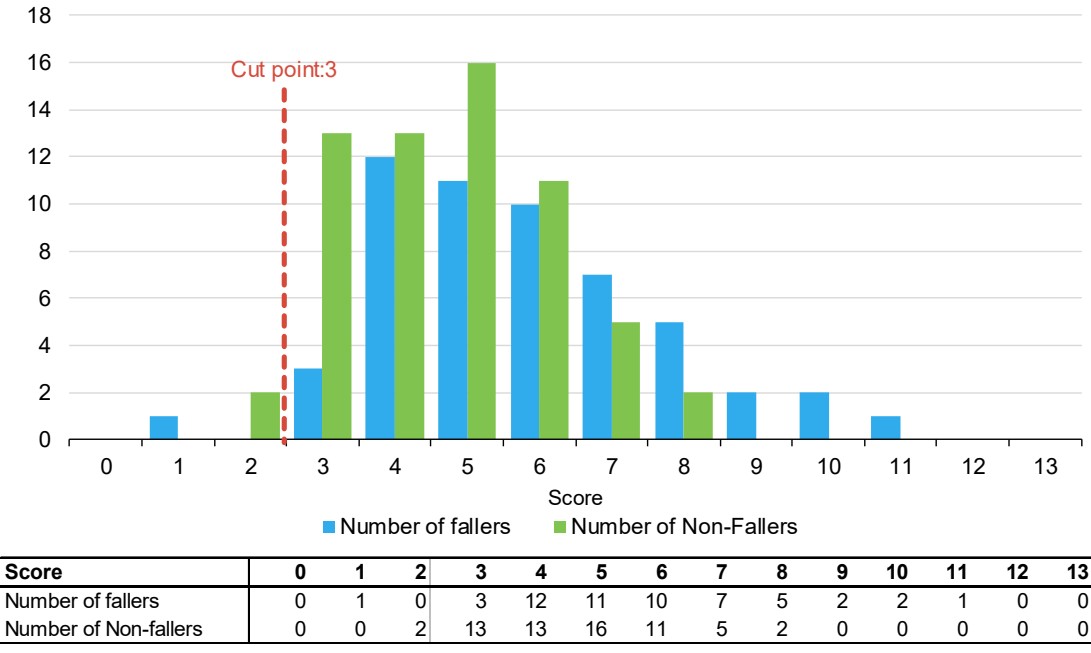

| Score | 0 | 1 | 2 | 3 | 4 | 5 | 6 | 7 | 8 | 9 | 10 | 11 | 12 | 13 |
|---|---|---|---|---|---|---|---|---|---|---|---|---|---|---|
| Number of fallers | 0 | 1 | 0 | 3 | 12 | 11 | 10 | 7 | 5 | 2 | 2 | 1 | 0 | 0 |
| Number of Non-fallers | 0 | 0 | 2 | 13 | 13 | 16 | 11 | 5 | 2 | 0 | 0 | 0 | 0 | 0 |

**Figure 3.** Number of fallers and non-fallers by WH-STRATIFY score.

The PH-FRAT was poor at differentiating the fall risks of participants. The majority (82%) of participants were assigned a high score of 16 which overcalls fallers, noting a cut point of 12 (Appendix B). In comparison, there is a larger spread of scores for TNH-STRATIFY and WH-STRATIFY (close to 90% of participants scored between 2 and 6 for TNH-STRATIFY and between 3 and 7 for WH-STRATIFY (Appendix B).

Of the three falls risk assessment tools using default cut points, TNH-STRATIFY has the highest predictive accuracy with a Youden Index[ER] of 0.20 and 95% Confidence Interval (CI) 0.07, 0.34. This difference was statistically significant (Table 2). PH-FRAT and WH-STRATIFY had similar predictive accuracy. PH-FRAT has a Youden Index[ER] 0.01 and 95% CI −0.04, 0.05. WH-STRATIFY has a Youden Index[ER] of 0.00 and 95% CI −0.04, 0.03

(Table 2). Both PH-FRAT and WH-STRATIFY had a sensitivity[ER] of 0.98 and a specificity[ER] of 0.02, respectively.

**Table 2.** Diagnostic predictive accuracy metrics for PH-FRAT, TNH-STRATIFY, and WH-STRATIFY using cut points for predicting a faller.

| Metric | PH-FRAT | | TNH-STRATIFY | | WH-STRATIFY | |
|---|---|---|---|---|---|---|
| Sensitivity[ER] | 0.98 | (0.95, 1.00) | 0.95 | (0.89, 1.00) | 0.98 | (0.94, 1.00) |
| Specificity[ER] | 0.02 | (0.00, 0.06) | 0.25 | (0.13, 0.38) | 0.02 | (0.00, 0.04) |
| Youden Index[ER] | 0.01 | (−0.04, 0.05) | 0.20 | (0.07, 0.34) | 0.00 | (−0.04, 0.03) |

Note—Values in parenthesis are bootstrapped 95% confidence intervals based on 1000 repetitions of the original sample size. The event rate metrics also factors for patients who may have had multiple falls in the same admission and the patient's length of stay [21].

### 2.3. Predictive Accuracy Using Optimal Cut Points

The default cut points for PH-FRAT and WH-STRATIFY were not optimal. The Youden Index[ER] can be maximised to 0.143 (at optimal cut point 17 for PH-FRAT) and 0.183 (at optimal cut point score 7 for WH-STRATIFY) (Table 3). The optimal cut point for TNH-STRATIFY is the default cut point (3).

**Table 3.** Event rate diagnostics by varying the fall cut-off score for PH-FRAT, TNH-STRATIFY, and WH-STRATIFY.

| Cut-Off Score | Sensitivity[ER] | Specificity[ER] | Youden Index[ER] |
|---|---|---|---|
| PH-FRAT | | | |
| 6 | 1.00 | 0.00 | 0.00 |
| 7 | 1.00 | 0.00 | 0.00 |
| 8 | 1.00 | 0.00 | 0.00 |
| 9 | 1.00 | 0.00 | 0.00 |
| 10 | 1.00 | 0.00 | 0.00 |
| 11 | 1.00 | 0.02 | 0.02 |
| 12 (Default) | 0.98 | 0.03 | 0.01 |
| 13 | 0.98 | 0.03 | 0.01 |
| 14 | 0.97 | 0.03 | −0.00 |
| 15 | 0.97 | 0.03 | −0.00 |
| 16 | 0.95 | 0.03 | −0.02 |
| 17 (Optimal) | 0.20 | 0.95 | 0.14 |
| 18 | 0.18 | 0.95 | 0.13 |
| 19 | 0.16 | 0.95 | 0.11 |
| 20 | 0.16 | 0.95 | 0.11 |
| TNH-STRATIFY | | | |
| 1 | 0.98 | 0.00 | −0.02 |
| 2 | 0.98 | 0.00 | −0.02 |
| 3 (Default, Optimal) | 0.95 | 0.25 | 0.20 |
| 4 | 0.66 | 0.53 | 0.19 |
| 5 | 0.39 | 0.72 | 0.11 |
| 6 | 0.30 | 0.87 | 0.17 |
| 7 | 0.20 | 1.00 | 0.20 |
| 8 | 0.12 | 1.00 | 0.12 |
| 9 | 0.07 | 1.00 | 0.07 |
| 10 | 0.00 | 1.00 | 0.00 |
| 11 | 0.00 | 1.00 | 0.00 |
| WH-STRATIFY | | | |
| 1 | 1.00 | 0.00 | 0.00 |
| 2 | 1.00 | 0.00 | 0.00 |
| 3 (Default) | 0.98 | 0.02 | 0.00 |

**Table 3.** *Cont.*

| Cut-Off Score | Sensitivity$^{ER}$ | Specificity$^{ER}$ | Youden Index$^{ER}$ |
|---|---|---|---|
| 4 | 0.93 | 0.22 | 0.15 |
| 5 | 0.69 | 0.40 | 0.09 |
| 6 | 0.49 | 0.69 | 0.18 |
| 7 (Optimal) | 0.33 | 0.86 | 0.18 |
| 8 | 0.21 | 0.96 | 0.18 |
| 9 | 0.13 | 1.00 | 0.13 |
| 10 | 0.08 | 1.00 | 0.08 |
| 11 | 0.05 | 1.00 | 0.05 |
| 12 | 0.00 | 1.00 | 0.00 |
| 13 | 0.00 | 1.00 | 0.00 |

PH-FRAT-Peninsula Health Falls Risk Screening Tool, TNH-STRATIFY-The Northern Hospital Modified St Thomas's Risk Assessment Tool, WH-STRATIFY-Western Health St. Thomas' Risk Assessment Tool (See Appendix C for graphs which illustrate these results).

Using optimal cut points, TNH-STRATIFY has the highest Youden Index$^{ER}$ (0.20) followed by WH-STRATIFY (0.18) and PH-FRAT (0.14) (Table 4). The Youden Index$^{ER}$ confidence intervals for PH-FRAT (0.01 to 0.29), WH-STRATIFY (0.00 to 0.35), and TNH-STRATIFY (0.07 to 0.34) overlap. Therefore, TNH-STRATIFY no longer has predictive accuracy superiority using optimal cut points, and the predictive accuracy for the three falls risk assessment tools is comparable.

**Table 4.** Diagnostic predictive accuracy metrics for PH-FRAT, TNH-STRATIFY, and WH-STRATIFY using optimal cut points for predicting a faller.

| Metric | PH-FRAT | | TNH-STRATIFY | | WH-STRATIFY | |
|---|---|---|---|---|---|---|
| Sensitivity$^{ER}$ | 0.20 | (0.09, 0.32) | 0.95 | (0.89, 1.00) | 0.33 | (0.19, 0.47) |
| Specificity$^{ER}$ | 0.95 | (0.87, 1.00) | 0.25 | (0.13, 0.38) | 0.86 | (0.72, 0.96) |
| Youden Index$^{ER}$ | 0.14 | (0.01, 0.29) | 0.20 | (0.07, 0.34) | 0.18 | (0.00, 0.35) |
| Sensitivity | 0.19 | (0.09, 0.29) | 0.94 | (0.88, 1.00) | 0.31 | (0.19, 0.44) |
| Specificity | 0.95 | (0.89, 1.00) | 0.26 | (0.16, 0.37) | 0.89 | (0.81, 0.95) |
| Youden Index | 0.14 | (0.02, 0.26) | 0.20 | (0.08, 0.33) | 0.20 | (0.06, 0.33) |

Values in parenthesis are bootstrapped 95% confidence intervals based on 1000 repetitions of the original sample size.

## 3. Discussion

This study showed that current tools have poor predictive accuracy for falls at their default cut points however, calculating optimal cut points for the three falls risk assessment tools shows promise for better predictive accuracy in high-risk inpatient populations. Using original cut points, PH-FRAT and WH-STRATIFY had no predictive accuracy for falls (Youden Index$^{ER}$ of 0.01 and 0.00 respectively). This study indicated that using WH-STRATIFY instead of PH-FRAT did not improve the predictive accuracy of falls on admission to GEM. Of the three falls risk assessment tools, PH-FRAT had the lowest falls prediction accuracy. These results are consistent with a previous large sample study on PH-FRAT [22]. This may be because the PH-FRAT assigns an automatic score of 16 when there is a change in function. As a change in function is a predominant reason for patients admitted to GEM, there is a selection bias that automatically classifies patients as high-risk for falls. In clinical practice, the low predictive accuracy of falls risk reduces the identification of patients at risk of falls thus misdirecting appropriate fall prevention strategies [6].

This study included the impact of adding two new risk factors (NESB and medications associated with falls) to TNH-STRATIFY to create a new falls risk assessment tool, WH-STRATIFY. These additions did not improve the predictive accuracy of WH-STRATIFY, in this study's population. Although the two additional risk factors were incorporated

in WH-STRATIFY, the default cut-point score remained at 3. We identified that adjusting this to 7 maximised predictive accuracy (Youden Index$^{ER}$ 0.00 at the default cut point 3, improved to Youden Index$^{ER}$ of 0.18 at optimal cut point 7). This remained lower than the Youden Index $^{ER}$ for TNH-STRATIFY at its optimal cut point (0.20), suggesting the addition of the two domains in WH-STRATIFY did not result in better predictive accuracy than TNH-STRATIFY. While there is evidence showing certain medications increase falls risk, there is limited evidence regarding whether patients with NESB have an increased risk of falls in hospital [23].

It is important to note there are risk factors for falls, such as sarcopenia and frailty, which are not captured in these risk assessment tools and should be considered in future.

The poor predictive accuracy overall suggests limitations with the use of numerical risk prediction tools in isolation for falls prediction and supports recommendations that clinical judgement and multi-disciplinary team-based assessment may be more effective than numerical risk prediction tools alone, in this setting. Falls risk assessment tools and management strategies should be locally designed according to the population and local resources available to improve efficacy. Calculating optimal cut points optimises the predictive accuracy of falls risk assessment tools to improve the identification of fall risk in clinical settings and allows for improved allocation of hospital resources targeting fall mitigation [23].

The optimal cut point for WH-STRATIFYreduces sensitivity and may not always be clinically desirable in a GEM population as this group is already at increased risk of falls, given reduced muscle strength and decline in mobility [1]. A carefully balanced consideration of competing factors is required to achieve effective fall prevention strategies by finding acceptable sensitivity and specificity ranges to reduce the misclassification of fallers or non-fallers. We have shown that falls risk assessment tools should undergo clinical validation and calculation of an optimal cut point before being incorporated into a fall prevention program [20].

When comparing the three falls risk assessment tools, TNH-STRATIFY demonstrated the best predictive accuracy with statistically significant and highest Youden Index $^{ER}$ of 0.20 (see Table 4), however, notably this tool's accuracy is still low to moderate at best. The tool's predictive accuracy at the optimal cut-point (Youden Index of 0.20), however, is also lower compared to the initial study where the calculated Youden Index for TNH-STRATIFY was 0.44 which suggests that TNH-STRATFY had a better efficacy when they are targeted to their local patient population [19]. It is common in falls risk assessment tool studies to show poor results replication [6,7,21,22]. This suggests that falls risk assessment tools' predictive accuracy may be affected by the patient sample and clinical setting. In other words, risk factors more predictive of fallers or non-fallers in the initial study setting are represented in the risk score, rather than in replicated studies [21,24]. Further studies are required to explore the underlying reasons for driving different predictive accuracies of falls risk assessment tools across different studies and whether this is due to certain changes in study population characteristics.

*Limitations of the Study*

This study had some strengths and limitations. This was an adaptation and validation study of a well-recognised risk assessment tool (TNH-STRATIFY) forming the WH-STRATIFY in a novel setting, as recommended in World Falls Guidelines [6]. In this retrospective study, only the numerical component of WH-STRATIFY could be assessed. The impact of WH-STRATIFY in its entirety requires future prospective evaluation of falls-risk management post-implementation to define how this tool performs in alignment with recent recommendations in favour of the use of multi-factorial risk assessment tools over numerical risk prediction tools [18].

Data depended on the quality of EMR documentation which was affected by incomplete data entry at the point of care. Due to the retrospective nature of this study, there was no opportunity to clinically assess patients in real-time during their admission or gather

any collateral history that would be helpful with calculating patients' falls risk beyond that which can be included in a numerical tool. When completing the falls risk assessment tools in clinical practice, staff may use alternate data sources rather than sole reliance on EMR data. However, the limitations of any missing data affected the collection of data for all three falls risk assessment tools similarly.

This study captures patients' falls risk on admission to GEM, however, in clinical practice, this risk may change during their admission which could alter the predictive ability of the tools. Regular re-scoring throughout a patient's admission using the WH-STRATIFY may lead to a more accurate score contemporaneous with the fall. Furthermore, the random selection of the non-faller comparator group showed non-significant differences in gender and age between fallers and non-fallers, and significant differences in length of stay. Prospective studies and feedback surveys from users of these tools at different time periods, including survival analysis for time-to-fall, could explore these limitations.

This study focused on a sample of GEM patients from different hospitals of a single health network which may reduce generalisability. This study minimised selection bias by randomisation and use of consistent criteria (Appendix A) when calculating the falls risk scores of each patient. Future studies across multiple hospital networks and patient groups in larger numbers would better determine generalisability of the findings.

## 4. Methods

### 4.1. Participants and Data Collection

We conducted this retrospective observational study at Western Health, a metropolitan health network servicing the Western Suburbs of Melbourne, Australia. This study involved patients admitted to four Geriatric Evaluation and Management (GEM) units across the network [6]. Participants were older adults, aged 65 or above, with acute deterioration in functional abilities due to illness, injury, or cognitive decline and were at risk of falls. This study focused on assessing numerical risk falls prediction tools so targeted interventions and prevention strategies could be implemented to mitigate fall risk during hospitalisation [6].

Adults aged 65 or above with inpatient falls admitted to GEM at Western Health between December 2020 and June 2021, identified through mandatory reporting, met inclusion criteria. This specific time window was selected to be outside the COVID-19 surge [25] to minimise changes to the typical patient profile and medical practices in GEM. A retrospective file review with data collection occurred in fifty-four consecutive patients who fell at least once during admission between December 2020 and June 2021. Sixty-two patients, aged 65 or above, admitted to GEM within the same time period who did not fall during their GEM admission were randomly selected for comparison in our study. We excluded any patients who were outside of the age of 65 or above and excluded any patients who were outside of the GEM admission period between December 2020 and June 2021. As this study was performed before the clinical launch of WH-STRATIFY only a retrospective assessment was possible.

Demographic and health data were manually retrieved from the Electronic Medical Record (EMR) between January 2022 and April 2022. The EMR was reviewed to score each participant with PH-FRAT, TNH-STRATIFY, and WH-STRATIFY (Appendix A). Data was stored in *REDCap*TM. Only data available on EMR on the day of and day preceding GEM admission were used. To reduce observation bias, blind scoring was conducted by randomising the total sample so knowledge of patients' fall status was not known by the assessor. As there are no previous studies of WH-STRATIFY, sample sizes could not be statistically determined *a priori*. The sample size of 116 was based on the sample size used in a comparable study [21].

### 4.2. Classification of Predicted Fallers and Non-Fallers

For prediction purposes, participants were classified as *predicted fallers* if they scored at or above the cut points of the falls risk assessment tools. Similarly, participants were classified as *predicted non-fallers* if they scored below the falls risk assessment tools' cut points.

### 4.3. Scoring of Falls Risk Assessment Tools

Using information available on the EMR on each patient's admission to GEM, the falls risk factors for these risk assessment tools (as listed in Table 5 below) were retrospectively identified and a score was calculated according to the three falls risk assessment tools' criteria and their cut-points.

**Table 5.** Summary of PH-FRAT, TNH-STRATIFY, and WH-STRATIFY.

| Tool | Scoring System | Risk Factor Assessed | Possible Score |
|---|---|---|---|
| PH-FRAT | Total score range: 5–20<br>Cut point: 12<br>Weighting of one risk factor (recent falls)<br>If having one either change in functional status/medication or dizziness/postural hypotension, the score is set to 16. If having both conditions, the score is set to 20. | Recent falls<br>Medication<br>Psychological<br>Cognitive status<br>Change in functional status/medication<br>Dizziness/postural hypotension | 2, 4, 6 or 8<br>1–4<br>1–4<br>1–4<br>16 or 20<br>16 or 20 |
| TNH-STRATIFY | Total score range: 0–11<br>Cut point: 3<br>Weighting of one risk factor (falls history—current admission) | Age<br>Falls history—current admission<br>Falls history—previous 12 months<br>Mental state<br>Mobility<br>Balance<br>Toileting needs<br>Vision impairment<br>Drug/alcohol abuse | 0 or 1<br>0 or 3<br>0 or 1<br>0 or 1<br>0 or 1<br>0 or 1<br>0 or 1<br>0 or 1<br>0 or 1 |
| WH-STRATIFY | Total score range: 0–13<br>Cut point: 3<br>Weighting of one risk factor (falls history—current admission) | Age<br>Falls history—current admission<br>Falls history—previous 12 months<br>Mental state<br>Mobility<br>Balance<br>Toileting needs<br>Vision impairment<br>Drug/alcohol abuse<br>NESB<br>Medications affecting mobility | 0 or 1<br>0 or 3<br>0 or 1<br>0 or 1<br>0 or 1<br>0 or 1<br>0 or 1<br>0 or 1<br>0 or 1<br>0 or 1<br>0 or 1 |

PH-FRAT—Peninsula Health Falls Risk Screening Tool, TNH-STRATIFY—The Northern Hospital Modified St Thomas's Risk Assessment Tool, WH-STRATIFY—Western Health St. Thomas' Risk Assessment Tool, NESB—Non-English Speaking Background. Please refer to Appendix A for further details of the falls risk assessment tools' scoring criteria.

The higher the participants scored according to the falls risk assessment tools, the higher likelihood of their fall risk as they are more likely to reach or exceed the tools' cut point. Participants with a score at or above the cut point were assessed by the falls risk assessment tool to be a predicted faller. The predictions were compared with the actual outcome during admission to assess each tool's predictive accuracy.

### 4.4. Ethics

Ethics approval was obtained from Western Health Office for Research (ERM ID 81444).

### 4.5. Statistical Analysis

Due to the variability of participants' falls' frequency and admission length, the fall *event rate* (*ER*) (defined as the frequency of falls during the patient's admission) was used [24]. *Sensitivity*$^{ER}$ is the number of falls during the patient's GEM admission correctly predicted by the falls risk assessment tool divided by the total number of falls. *Specificity*$^{ER}$ is the length of hospital stay for non-fallers accurately predicted to have low fall risk by the falls risk assessment tool divided by the total length of hospital stay for all non-fallers. The event rate *Youden Index* (*Youden Index*$^{ER}$) [6,24] measured the predictive accuracy of the falls risk assessment tools. The *Youden Index*$^{ER}$ is the sum of the event rate sensitivity (sensitivity$^{ER}$) and event rate specificity (specificity$^{ER}$) less 1 and produces a value between −1 and 1, with a higher value indicating greater predictive accuracy.

The *Youden Index*$^{ER}$ is preferred [24,26] as it provides a measure of accuracy by equally weighing sensitivity (i.e., the tool correctly predicts the patient is at high risk of fall) and specificity (i.e., the tool correctly predicts the patient is at low risk of fall). It adjusts for patients who had multiple falls and GEM length of stay. Statistical significance was assessed using bootstrapped 95% confidence intervals. *Bootstrapping* is a statistical method that resamples a single data set of the current study to create many simulated samples. This process allows the construction of the confidence intervals [6] and other studies have also used this method to derive 95% confidence intervals for the Youden Index$^{ER}$ [26]. If the 95% confidence intervals for two falls risk assessment tools overlap, this implies that the two tools do not differ significantly at a 5% level.

The optimal cut point is the score that maximises the predictive accuracy of the three falls risk assessment tools (WH-STRATIFY, PH-FRAT, and TNH-STRATIFY). Using optimal cut points reduces misclassification of those who are likely or not to fall during GEM admission. Microsoft Excel was used in the statistical implementation.

### 4.6. Deriving Optimal Cut Points

Cut points were modified to demonstrate impact on the sensitivity$^{ER}$, specificity$^{ER}$, and Youden Index$^{ER}$. We selected the optimal cut point as the score where the Youden Index$^{ER}$ is at its highest calculated value.

## 5. Conclusions

Of the three falls risk assessment tools (TNH-STRATIFY, WH-STRATIFY, and PH-FRAT) using the default cut points, TNH-STRATIFY offered the highest predictive accuracy. PH-FRAT and WH-STRATIFY at default cut points had negligible predictive accuracy, and showed modest improvement when using calculated optimal cut points. However, this study showed the predictive accuracy for all three falls risk assessment tools remained low at their default and optimal cut-points.

This study supports the recommendation that numerical risk prediction tools in isolation are insufficient for the purpose of identifying patients at high risk of falls and guiding falls risk interventions Clinical judgment and multi-disciplinary assessments are important to help predict the falls risk for inpatients and enable the use of person-centered falls risk reduction strategies [16]. Future prospective research is required to assess the utility of WH-STRATIFY and associated individualised fall risk interventions as part of a falls risk management program [16]. The optimal cut points of falls risk assessment tools should be determined and validated in different clinical settings to optimise predictive accuracy, support targeted falls risk mitigation, and improve resource allocation.

**Author Contributions:** V.L.: Conceptualisation, Methodology, Formal analysis, Investigation, Writing-Original Draft, Review & Editing, Project administration. J.M.: Conceptualisation, Methodology, Writing—Review & Editing, Formal Analysis, Supervision. L.A.-K.: Conceptualisation, Methodology, Writing—Review & Editing, Formal Analysis, Supervision. J.Z.: Writing—Review & Editing, Formal analysis, Supervision. S.V.: Formal analysis. All authors have read and agreed to the published version of the manuscript.

**Funding:** This research received no external funding.

**Institutional Review Board Statement:** Ethics approval was obtained from Western Health Office for Research (ERM ID 81444).

**Informed Consent Statement:** Not applicable.

**Data Availability Statement:** Data is unavailable due to privacy or ethical restrictions.

**Conflicts of Interest:** The authors declare no conflict of interest.

## Appendix A Falls Risk Assessment Tool Data Definition

Table A1 shows the data collection table used to collect all of the patient data and calculate a score with respect to each of the three falls risk assessment tools.

**Table A1.** Falls Risk Assessment Tool data definition.

| Questions/Risk Factors | Descriptions |
|---|---|
| 1. Demographics<br>Age (years)<br>Gender: Female, Male | Source: (including the time frame of when the search is for)<br>• Electronic Medical Record (EMR) Patient information on admission<br>Definition:<br>• Age (rounded to the nearest whole year)<br>• Gender—what is documented on patient information on EMR |
| 2. Is the patient aged 80 or older?<br>Yes/No | Source: (including the time frame of when the search is for)<br>• EMR Patient information on admission<br>Definition:<br>Age (to the whole year) |
| 3. Length of stay<br>Days | Source: (including the time frame of when the search is for)<br>• EMR inpatient recorded the date of admission to and date of discharge from GEM<br>Definition:<br>• Length of inpatient stay from day of admission to the day of discharge from GEM to any other location (including home, residential care, or another inpatient unit)<br>• Rounded to the nearest whole day |
| 4. Was there a recent fall?<br>• None in the last 12 months One or more<br>• Within the last 0–3 months, Within the last 3–12 months, the Patient is in the hospital primarily due to a fall<br>• Fall during current admission (interval from the day of admission to the day of fall (in days) | Source: (including the time frame of when the search is for)<br>• EMR Medical GEM admission note (including any past medical history of falls or if fall was a reason for admission to hospital), nursing admission note<br>• EMR Initial patient assessment completed by nursing on admission<br>• EMR Adults risk assessment on the day of admission to GEM à Falls Assessment<br>• EMR date of admission and date of patient's fall in GEM<br>• RiskMan data to correlate patients who fell in GEM at Williamstown, Footscray, and Sunshine Hospitals from February to May 2021<br>Definition:<br>Search for keywords "Fall" or "Falls" on EMR on the patient's GEM admission |

**Table A1.** *Cont.*

| Questions/Risk Factors | Descriptions |
|---|---|
| 5. Is the patient taking the following medications—sedatives, antidepressants, anti-parkinson's, diuretics, anti-hypertensives, hypnotics, or opioids? No/Taking one/Taking two/Taking more than two<br><br>Class(es): Sedatives, Antidepressant, Anti-Parkinson's, Diuretics, Anti-hypertensives, Hypnotics, Opioids | Source: (including the time frame of when the search is for)<br>• Medical admission medication list<br>• Medication Administration Record (MAR) medications charted on the day of admission to GEM<br>• Pharmacy medication reconciliation list on admission to GEM<br><br>Definition:<br>Medication classes that are charted or documented on admission to the GEM unit including both regular and as-required medications:<br>• Sedatives<br>• Anti-depressants<br>• Anti-Parkinson's<br>• Diuretics<br>• Hypnotics<br>• Opioids |
| 6. Is the patient affected by psychological conditions including: anxiety, depression, reduced cooperation, reduced insight, or reduced judgment?<br>• No<br>• Appears mildly affected by one or more<br>• Appears moderately affected by one or more<br>• Appears severely affected by one or more | Source: (including the time frame of when the search is for)<br>• EMR Behaviours of concern<br>• EMR Mental Status—Orientation, Affect/Behaviour, Hallucinations present<br>• EMR Medical GEM admission note (including any past medical history of psychological condition<br>• EMR Interactive view → Adult Systems Assessment à Mental Status → Affect/Behaviour, Hallucinations present<br><br>Source: (including the time frame of when the search is for)<br>• EMR Initial patient assessment completed by nursing on admission<br>• EMR Adults risk assessment on admission to GEM à Falls Assessment<br>• EMR date of admission and date of patient's fall in GEM<br><br>Definition:<br>• EMR search for keywords including:<br>  a. Depression<br>  b. Anxiety<br>  c. Feeling down<br>  d. Affect<br>  e. Reduced insight<br>  f. Poor judgment<br>• Grading severity<br>  a. Mildly affected<br>  b. Psychological condition has not impacted on daily ADLs<br>  c. No change to antipsychotic or antidepressant on the GEM admission medication list<br>  d. Moderately affected<br>  e. Psychological condition is an active issue as per the medical team on GEM admission<br>  f. As required antipsychotic or benzodiazepine on the GEM admission medication list<br>  g. Severely affected<br>  h. Psychological condition is the reason for the patient's acute hospitalisation prior to transfer to GEM<br>  i. Code grey or requiring IM antipsychotic<br><br>Psychological condition led to the patient needing a 1:1 special on admission to GEM |

**Table A1.** *Cont.*

| Questions/Risk Factors | Descriptions |
|---|---|
| 7. What is the patient's cognition?<br>• Cognitively intact<br>• Mildly cognitive impairment<br><br>If mildly cognitively impaired, does it include any of:<br>• Confused<br>• Impulsive<br>• Agitated<br>• Moderate cognitive impairment<br>• Severe cognitive impairment<br><br>Note: Moderate to severe cognitive impairment will already automatically include confused, impulsive, and/or agitated behaviours. | Source: (including the time frame of when the search is for)<br>• EMR Medical GEM admission note<br>• EMR nursing admission note<br>• EMR OT admission note<br>• EMR Initial patient assessment à Premorbid Information<br>• EMR Interactive View à Adult Risk assessment à Cognition—Abbreviated Mental Test (4AT-Delirium screening test) on admission<br>Definition:<br>• EMR search for keywords including:<br>  a. Dementia<br>  b. Delirium<br>  c. Cognitive impairment<br>• Mild cognitive impairment<br>  a. Cognitive impairment has not affected independence with Activities of Daily Living (ADLs), or<br>  b. 4AT score 1–3 (delirium screening test)<br>• Moderate cognitive impairment<br>  a. Cognitive impairment has led to needing assistance with domestic ADLs from others, or<br>  b. 4AT score 4–7 (Abbreviated Mental Test delirium screening test)<br>• Severe cognitive impairment<br>  a. Requires full assistance with ADLs, or<br>  b. 4AT score 8–12 |
| 8. What is the patient's level of mobility? supervision or assistance when mobilising?<br>• Independent<br>• Supervision<br>• Assistance<br>• Impaired balance<br>• Hemiplegia<br>• Gait aid | Source: (including the time frame of when the search is for)<br>• EMR Medical GEM admission note<br>• EMR nursing admission note<br>• EMR Physiotherapist admission note<br>• EMR Initial patient assessment à premorbid information completed by nursing on admission<br>• EMR Interactive view → Adult Systems Assessment → Neurological Gait, Upper limb movement/Strength, Lower limb movement/Strength<br>Definition:<br>• Review documentation to see if, on admission to GEM, the patient needs supervision or assistance (a person to help i.e., not just mobility aid) when mobilising<br>• Search for words on the patient's GEM medical admission including:<br>  a. Impaired balance or poor balance or balance issues<br>  b. Unsteadiness or unsteady<br>Hemiplegia—unilateral arm +/− leg weakness |

**Table A1.** *Cont.*

| Questions/Risk Factors | Descriptions |
|---|---|
| 9. Does the patient require frequent toileting of bowels +/− bladder? Yes/No | Source: (including the time frame of when the search is for) <br>• EMR Medical GEM admission note, nursing admission note <br>• EMR Initial patient assessment completed by nursing on admission <br>• EMR Adults risk assessment on admission to GEM (nursing note) → Continence Assessment → Incontinent of Urine, Incontinent of faeces, Urgency <br>• EMR Interactive view à Adult Systems Assessment → Activities of Daily Living—Hygiene—Incontinence Aid <br><br>Definition: <br>• Review the nursing and medical notes on the patient's admission about urgency, urinary incontinence, and bowel incontinence <br>• Look for terms on GEM admission notes including: <br>   a. Urinary incontinence <br>   b. Faecal incontinence <br>   c. Incontinent <br><br>Urgency (difficulty toileting in time) |
| 10. Does the patient have vision impairment which affects everyday functioning? Yes/No | Source: (including the time frame of when the search is for) <br>• EMR Medical GEM admission note <br>• EMR nursing admission note <br>• EMR Initial patient assessment completed by nursing on admission à look at Speech and Sensory deficits and check for Visual impairment, glasses/contact lenses <br>• EMR Interactive View → Adult Systems Assessment → Mobility status <br><br>Definition: <br>• On the patient's GEM admission, does the patient have/described the following: <br>   a. Glasses <br>   b. Contact lenses <br>   c. Visually impaired <br><br>Legally blind |
| 11. Does the patient present with drug/alcohol-related issues? Yes/No | Source: (including the time frame of when the search is for) <br>• EMR Initial patient assessment completed by nursing on admission → Social history → Substance Use, Alcohol <br>• EMR Medical GEM admission note social history <br>• EMR nursing admission note <br><br>Definition: <br>• On the patient's admission to GEM, look for keywords including: <br>   a. Illicit Drugs <br>   b. Recreational drugs <br>   c. Substance Use <br>   d. Drug abuse <br>   e. Addiction (relating to illicit drugs and/or alcohol) <br>   f. Alcohol <br>   g. Etoh <br>   h. AWS (Alcohol Withdrawal Scale) <br><br>Alcohol withdrawal |
| 12. Does the patient require a language interpreter? Yes/No | Source: (including the time frame of when the search is for) <br>• EMR Patient information → Interpreter Required <br>• EMR Medical GEM admission notes <br>• EMR Nursing GEM admission notes <br><br>Definition: <br>EMR search term for "Interpreter" |

**Table A1.** *Cont.*

| Questions/Risk Factors | Descriptions |
|---|---|
| 13. Does the patient have any recent change in functional status or medications that affect the safety of mobility?<br>● No<br>● Yes<br>● Medications<br>● Functional status change | Source: (including the time frame of when the search is for)<br>● EMR Medical GEM admission notes<br>● EMR Nursing GEM admission notes<br>Definition:<br>● Medication that had led to sedation or affected mobility including dizziness, postural instability, and postural hypotension.<br>Change in functional status that led to change to safety with mobility including needing a new gait aid, needing supervision with mobility/transfers, needing assistance with mobility/transfers |
| 14. Does the patient experience dizziness or postural hypotension?<br>● No<br>● Yes<br>● Dizziness<br>● Postural hypotension | Source: (including the time frame of when the search is for)<br>● EMR Medical GEM admission notes<br>● EMR Nursing GEM admission notes<br>Definition:<br>● Terms including: "Dizziness" or "Dizzy"<br>● Postural BP on the day of admission or documented on the patient's GEM admission note that it occurred during acute admission.<br>Significant postural hypotension is a postural drop of at least 20 mmHg systolic or 10 mmHg diastolic from a sitting or supine position to standing |

| PH-FRAT score<br>Low Risk: 5–11<br>Medium: Risk: 12–15<br>High Risk: 16–20<br><br>A score of 12 or above suggests an increased risk of falls<br><br>Automatic High risk<br>A recent change in functional status and/or medications affecting the safety of mobility (or anticipated)<br>Dizziness/postural hypotension | Risk factor | Level | Risk score |
|---|---|---|---|
| | Recent Falls<br>(To score this, complete history of falls, overleaf) | none in the last 12 months | 2 |
| | | one or more between 3 and 12 months ago | 4 |
| | | one or more in the last 3 months | 6 |
| | | one or more in the last 3 months whilst inpatient/resident | 8 |
| | Medications<br>(Sedatives, Anti-Depressants Anti-Parkinson's, Diuretics Anti-hypertensives, hypnotics) | not taking any of these | 1 |
| | | taking one | 2 |
| | | taking two | 3 |
| | | taking more than two | 4 |
| | Psychological<br>(Anxiety, Depression Cooperation, Insight or Judgement esp. re mobility) | does not appear to have any of these | 1 |
| | | appears mildly affected by one or more | 2 |
| | | appears moderately affected by one or more | 3 |
| | | appears severely affected by one or more | 4 |
| | Cognitive status<br>(AMTS: Hodkinson Abbreviated Mental Test Score) | AMTS 9 or 10/10 OR intact | 1 |
| | | AMTS 7–8 mildly impaired | 2 |
| | | AMTS 5–6 moderately impaired | 3 |
| | | AMTS 4 or less severely impaired | 4 |
| | Risk Score (Low Risk: 5–11 Medium: Risk: 12–15 High Risk: 16–20) | | /20 |
| | Automatic High-Risk Status: (if ticked then circle HIGH risk below) | | |
| | ● Recent change in functional status and/or medications affecting safe mobility (or anticipated)<br>● Dizziness/postural hypotension | | |

**Table A1.** *Cont.*

| Questions/Risk Factors | Descriptions | |
|---|---|---|
| | Risk factors | Risk score |
| | Fall: During current Admission | Yes = 3, No = 0 |
| | Fall: Within 12 months | Yes = 1, No = 0 |
| NH-STRATIFY score A score of 3 or more is considered High Risk | Mental State-Current cognition—confused, impulsive, agitated, or cognitively impaired | Yes = 1, No = 0 |
| | Mobility: Patient needs supervision or assistance when mobilising | Yes = 1, No = 0 |
| | Impaired Balance and/or hemiplegia | Yes = 1, No = 0 |
| | Age 80 or older | Yes = 1, No = 0 |
| | Frequent toileting bowels +/− bladder | Yes = 1, No = 0 |
| | Vision impairment—that affects everyday functioning | Yes = 1, No = 0 |
| | Drug and alcohol: patient presents with drug/alcohol-related issues | Yes = 1, No = 0 |
| | Risk factors | Risk score |
| | Fall: During current Admission | Yes = 3, No = 0 |
| | Fall: Within 12 months | Yes = 1, No = 0 |
| | Language: patient is NESB | Yes = 1, No = 0 |
| | Current cognition—confused, impulsive, agitated, or cognitively impaired | Yes = 1, No = 0 |
| WH-STRATIFY score A score of 3 or above is considered High falls risk | Vision impairment—that affects everyday functioning | Yes = 1, No = 0 |
| | Mobility: Patient needs supervision or assistance when mobilising | Yes = 1, No = 0 |
| | Impaired Balance and/or hemiplegia | Yes = 1, No = 0 |
| | Age 80 or older | Yes = 1, No = 0 |
| | Frequent toileting bowels +/− bladder | Yes = 1, No = 0 |
| | Medications affecting mobility: anti-hypertensives, diuretics, sedatives, opioids or S11 | Yes = 1, No = 0 |
| | Drug and alcohol: patient presents with drug/alcohol-related issues | Yes = 1, No = 0 |

## Appendix B  Summary Statistics by Falls Risk Assessment Tool

Table A2 compared three falls risk assessment tools (PH-FRAT, TNH-STRATIFY, and WH-STRATIFY) and looked at the number of patients who were classified into each of the numerical scores according to each tool. It also looked at patients' length of stay, the number of patients who fell, and the total number of falls.

**Table A2.** The number of patients, fallers, patient days, and falls by scores produced under PH-FRAT, TNH-STRATIFY, and WH-STRATIFY.

| Score | Number of Patients (%) | | Number of Fallers (%) | | Length of Stay, Days (%) | | Number of Falls | |
|---|---|---|---|---|---|---|---|---|
| PH-FRAT | | | | | | | | |
| 5 | 0 | (0%) | 0 | (0%) | 0 | (0%) | 0 | (0%) |
| 6 | 0 | (0%) | 0 | (0%) | 0 | (0%) | 0 | (0%) |
| 7 | 1 | (1%) | 0 | (0%) | 2 | (0%) | 0 | (0%) |
| 8 | 0 | (0%) | 0 | (0%) | 0 | (0%) | 0 | (0%) |
| 9 | 0 | (0%) | 0 | (0%) | 0 | (0%) | 0 | (0%) |
| 10 | 2 | (2%) | 0 | (0%) | 13 | (1%) | 0 | (0%) |
| 11 | 2 | (2%) | 1 | (2%) | 36 | (2%) | 1 | (2%) |
| 12 | 1 | (1%) | 0 | (0%) | 5 | (0%) | 0 | (0%) |
| 13 | 1 | (1%) | 1 | (2%) | 31 | (1%) | 1 | (2%) |
| 14 | 0 | (0%) | 0 | (0%) | 0 | (0%) | 0 | (0%) |
| 15 | 1 | (1%) | 1 | (2%) | 24 | (1%) | 1 | (2%) |
| 16 | 95 | (82%) | 41 | (76%) | 1896 | (80%) | 46 | (75%) |
| 17 | 1 | (1%) | 1 | (2%) | 56 | (2%) | 1 | (2%) |
| 18 | 1 | (1%) | 1 | (2%) | 32 | (1%) | 1 | (2%) |

**Table A2.** *Cont.*

| Score | Number of Patients (%) | | Number of Fallers (%) | | Length of Stay, Days (%) | | Number of Falls | |
|---|---|---|---|---|---|---|---|---|
| 19 | 0 | (0%) | 0 | (0%) | 0 | (0%) | 0 | (0%) |
| 20 | 11 | (9%) | 8 | (15%) | 264 | (11%) | 10 | (16%) |
| TNH-STRATIFY | | | | | | | | |
| 0 | 1 | (1%) | 1 | (2%) | 35 | (1%) | 1 | (2%) |
| 1 | 3 | (3%) | 1 | (2%) | 31 | (1%) | 1 | (2%) |
| 2 | 15 | (13%) | 1 | (2%) | 205 | (9%) | 1 | (2%) |
| 3 | 34 | (29%) | 15 | (28%) | 624 | (26%) | 18 | (30%) |
| 4 | 26 | (22%) | 15 | (28%) | 722 | (31%) | 16 | (26%) |
| 5 | 16 | (14%) | 6 | (11%) | 247 | (10%) | 6 | (10%) |
| 6 | 12 | (10%) | 6 | (11%) | 274 | (12%) | 6 | (10%) |
| 7 | 4 | (3%) | 4 | (7%) | 89 | (4%) | 5 | (8%) |
| 8 | 3 | (3%) | 3 | (6%) | 65 | (3%) | 3 | (5%) |
| 9 | 2 | (2%) | 2 | (4%) | 67 | (3%) | 4 | (7%) |
| 10 | 0 | (0%) | 0 | (0%) | 0 | (0%) | 0 | (0%) |
| 11 | 0 | (0%) | 0 | (0%) | 0 | (0%) | 0 | (0%) |
| WH-STRATIFY | | | | | | | | |
| 0 | 0 | (0%) | 0 | (0%) | 0 | (0%) | 0 | (0%) |
| 1 | 1 | (1%) | 1 | (2%) | 35 | (1%) | 1 | (2%) |
| 2 | 2 | (2%) | 0 | (0%) | 14 | (1%) | 0 | (0%) |
| 3 | 16 | (14%) | 3 | (6%) | 222 | (9%) | 3 | (5%) |
| 4 | 25 | (22%) | 12 | (22%) | 469 | (20%) | 15 | (25%) |
| 5 | 27 | (23%) | 11 | (20%) | 587 | (25%) | 12 | (20%) |
| 6 | 21 | (18%) | 10 | (19%) | 461 | (20%) | 10 | (16%) |
| 7 | 12 | (10%) | 7 | (13%) | 307 | (13%) | 7 | (11%) |
| 8 | 7 | (6%) | 5 | (9%) | 134 | (6%) | 5 | (8%) |
| 9 | 2 | (2%) | 2 | (4%) | 43 | (2%) | 3 | (5%) |
| 10 | 2 | (2%) | 2 | (4%) | 44 | (2%) | 2 | (3%) |
| 11 | 1 | (1%) | 1 | (2%) | 43 | (2%) | 3 | (5%) |
| 12 | 0 | (0%) | 0 | (0%) | 0 | (0%) | 0 | (0%) |
| 13 | 0 | (0%) | 0 | (0%) | 0 | (0%) | 0 | (0%) |

Percentages in parenthesis represent the proportion of the column total for each falls risk assessment tool.

Table A3 and Figure A1 show the impact on the predictive accuracy diagnostics by varying the falls cut-off score. This table compared the three falls risk assessment tools (PH-FRAT, WH-STRATIFY, and TNH-STRATIFY) with regards to their cut-off scores demonstrated their respective Youden Index[ER]

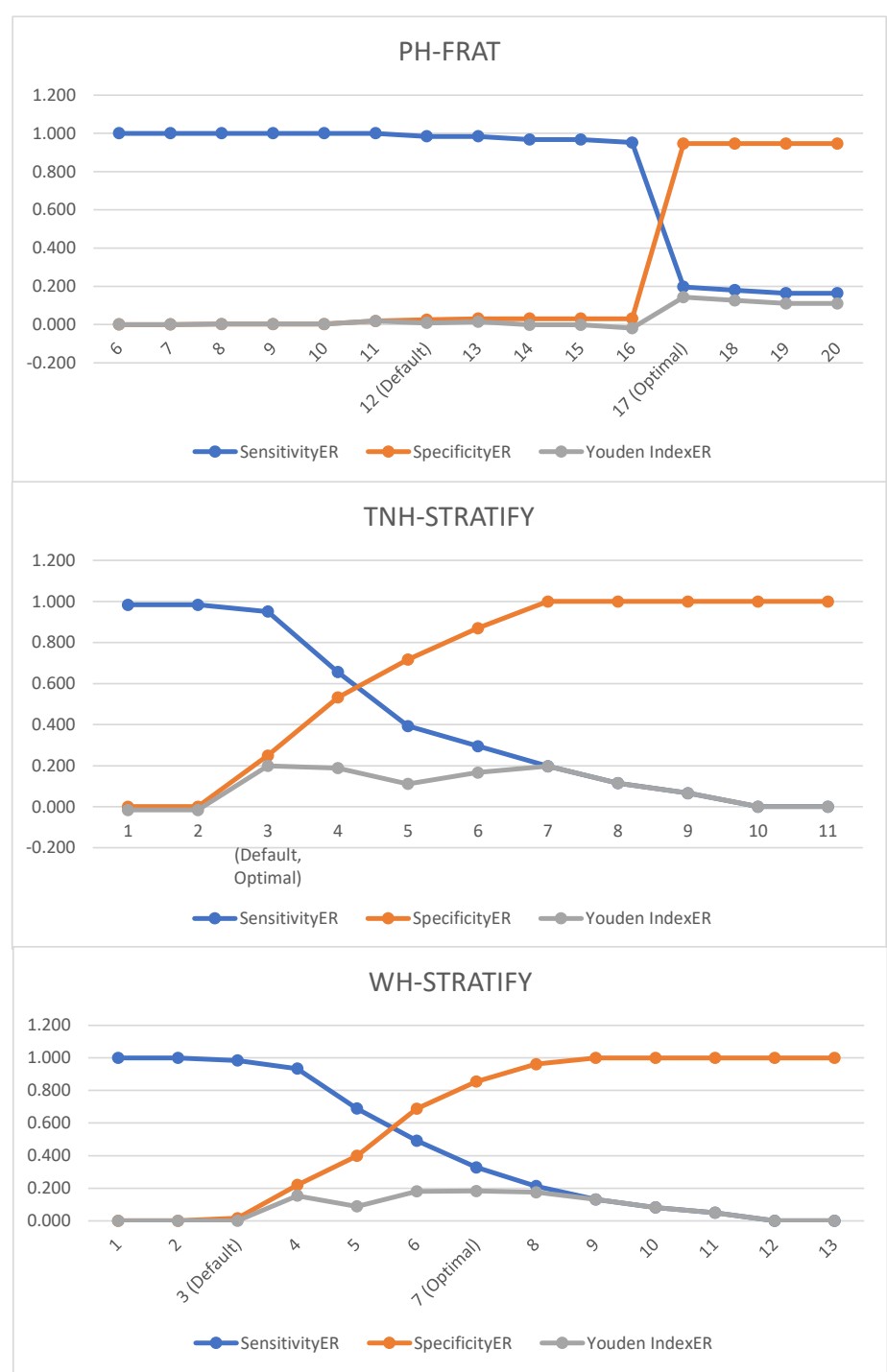

**Figure A1.** Graph of event rate diagnostics by varying the fall cut-off score for PH-FRAT, TNH-STRATIFY, and WH-STRATIFY.

**Table A3.** Event rate diagnostics by varying the fall cut-off score for PH-FRAT, TNH-STRATIFY, and WH-STRATIFY.

| Cut-Off Score | Sensitivity[ER] | Specificity[ER] | Youden Index[ER] |
|---|---|---|---|
| PH-FRAT | | | |
| 6 | 1.00 | 0.00 | 0.00 |
| 7 | 1.00 | 0.00 | 0.00 |
| 8 | 1.00 | 0.00 | 0.00 |

**Table A3.** *Cont.*

| Cut-Off Score | Sensitivity[ER] | Specificity[ER] | Youden Index[ER] |
|---|---|---|---|
| 9 | 1.00 | 0.00 | 0.00 |
| 10 | 1.00 | 0.00 | 0.00 |
| 11 | 1.00 | 0.02 | 0.02 |
| 12 (Cut-off) | 0.98 | 0.03 | 0.01 |
| 13 | 0.98 | 0.03 | 0.01 |
| 14 | 0.97 | 0.03 | −0.00 |
| 15 | 0.97 | 0.03 | −0.00 |
| 16 | 0.95 | 0.03 | −0.02 |
| 17 | 0.20 | 0.95 | 0.14 |
| 18 | 0.18 | 0.95 | 0.13 |
| 19 | 0.16 | 0.95 | 0.11 |
| 20 | 0.16 | 0.95 | 0.11 |
| TNH-STRATIFY | | | |
| 1 | 0.98 | 0.00 | −0.02 |
| 2 | 0.98 | 0.00 | −0.02 |
| 3 (Cut-off) | 0.95 | 0.25 | 0.20 |
| 4 | 0.66 | 0.53 | 0.19 |
| 5 | 0.39 | 0.72 | 0.11 |
| 6 | 0.30 | 0.87 | 0.17 |
| 7 | 0.20 | 1.00 | 0.20 |
| 8 | 0.12 | 1.00 | 0.16 |
| 9 | 0.07 | 1.00 | 0.07 |
| 10 | 0.00 | 1.00 | 0.00 |
| 11 | 0.00 | 1.00 | 0.00 |
| WH-STRATIFY | | | |
| 1 | 1.00 | 0.00 | 0.00 |
| 2 | 1.00 | 0.00 | 0.00 |
| 3 (Cut-off) | 0.98 | 0.02 | 0.00 |
| 4 | 0.93 | 0.22 | 0.15 |
| 5 | 0.69 | 0.40 | 0.09 |
| 6 | 0.49 | 0.69 | 0.18 |
| 7 | 0.33 | 0.86 | 0.18 |
| 8 | 0.21 | 0.96 | 0.18 |
| 9 | 0.13 | 1.00 | 0.13 |
| 10 | 0.08 | 1.00 | 0.08 |
| 11 | 0.05 | 1.00 | 0.05 |
| 12 | 0.00 | 1.00 | 0.00 |
| 13 | 0.00 | 1.00 | 0.00 |

## Appendix C  WH STRATIFY Interventions

Table A4 shows the suggested interventions for the WH STRATIFY.

**Table A4.** WH STRATIFY falls risk assessment tool and its suggested interventions.

| Questions | Optional Answer | Optional Answer | If Yes—Drop Down Multi Choice Options |
|---|---|---|---|
| Fall: Current Admission | No (score 0) | Yes patient has had a fall during current admission (score 3) | Bathroom Supervision at all times<br>Ensure physiotherapy is involved in the care<br>Consider referral to OT<br>Handover History of falls to oncoming shift<br>Monitor postural BP for 48/24. Report drop of 20 mmHg |
| Fall: Within 12 months | No (0) | Yes patient has had fall/s in the last 12 months (1) | Provide falls prevention education<br>Monitor postural BP for 48/24. Report drop of 20 mmHg |

**Table A4.** *Cont.*

| Questions | Optional Answer | Optional Answer | If Yes—Drop Down Multi Choice Options |
|---|---|---|---|
| Language: | Speaks and understands English | Patient does not speak or understand English (1) | Phone interpreter-falls education/orientation/4AT Write common word translations on the patient whiteboard Ask the family to assist with orientation/falls education |
| Current cognition | No Cognitive impairment (0) | Confused, impulsive, agitated, or cognitively impaired (1) | Lolo bed with crash mats Bathroom supervision at all times Falls Mat alarm Locate the patient closer to the nurses' station Portable Video Monitoring Overnight Request family stay with the patient Assess for constipation/overflow/bowel sounds Complete 4AT and report a score of 4+ to HMO Pain assessment Toilet regime pre + post meals Review the About Me form Mobilise regularly Update patient whiteboard each shift |
| Vision impairment | No Visual impairment (0) | Yes visual impairment that affects everyday functioning (1) | Vision impaired sign above the bed Consider using a manual handbell Ensure glasses are within reach Co-locate with other patients if suitable |
| Mobility Impaired | No Mobility Impairment (0) | Yes patient needs supervision or assistance when mobilising (1) | Refer to Physiotherapy if change in baseline function Ensure gait aid is within reach at all times |
| Impaired Balance: | No Balance Impairment (0) | Yes Patient has impaired balance and/or hemiplegia (1) | Refer to Physiotherapy Reinforce PT mobility instructions Ensure gait aid is within reach at all times Bathroom supervision at all times |
| Age, over 80: | No (0) | Yes patient is 80 years or older (1) | Educate on increased falls risk |
| Frequent toileting/urgency: | No (0) | yes patient requires frequent toileting: bladder +/− bowels (1) | Bedside commode or access to bottle Consider proximity to the toilet Check urine—FWT or MSU Toileting regime Monitor for constipation and overflow Educate on suitable continence aids Consider a bladder scan for retention |
| Medications affecting mobility: | One or none (0) | 2 or more: antihypertensives, diuretics, sedatives, opioids, or S11 (1) | Educate patient—increased falls risk due to meds |
| Drug and alcohol issues | No (0) | Yes patient presents with drug/alcohol-related issues (1) | Consider referral to addiction medicine |
| Patient falls risk: | Low Risk = less than 3 | High Risk—3 or more | |
| Standard Falls risk strategies—All Patients | | Focus is on identifying risk factors and implementing prevention strategies regardless of risk rating. | Discuss strategies to keep patients safe in the hospital Reduce all clutter Call bell always within reach Gait aid is always within reach Use non-slip footwear during daytime—not socks Dress in day clothes if available |

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
