# Peer review of "Current Cut Points of Three Falls Risk Assessment Tools Are Inferior to Calculated Cut Points in Geriatric Evaluation and Management Units"

_muscles, doi:10.3390/muscles2030019_

Round 1
Reviewer 1 Report
Thank you for inviting me to review this manuscript. It is an interesting study exploring the predictive accuracy of fall risk assessment tools. The authors should be congratulated on conducting this experiment and using this specific research design, however to improve the manuscript I would suggest the following:
ABSTRACT
As participants were older adults, I would suggest the authors to provide a more specific title to include this information too.
The abstract is generally well written, however it would be helpful to specify what is different in the more recently adopted assessment tool compared to the previous one. This will help making the rationale for the current study stronger.
INTRODUCTION
I would suggest the authors to provide a definition of fall to provide a full description of the context. Also, it is not clear why (line 54-55) the authors focus mainly on sarcopenia in the first paragraph.
Please make sure to write in paragraphs.
Information included in line 66-73 is more appropriate for the methods section. I would suggest the authors trying to focus more on introducing why is important to study the topic and what is the novelty of this study.
Please make sure to clearly state the hypothesis of this study.
METHODS
The authors should please provide more details on inclusion and exclusion criteria.
Please provide a supporting reference for the use of the fall event rate (line 289).
Please provide details on how data were presented and on demographics collected too.
Table 5, I would suggest adding information on score interpretation i.e. higher score higher risk of falls..
RESULTS
I have some doubts on the decision of randomly selecting for the non-fallers group. Particularly as the non-faller group had a significant longer average admission period. Could the authors please justify with more details this decision? Why not matching the two groups? Also, the gender ratio looks quite imbalanced, this is important due to potential musculoskeletal differences and sarcopenia rates too.
The authors are occasionally referring to sarcopenia, however there is no mention of assessing it in the methods section. Please make sure to include only relevant information.
Please make sure to be consistent with the number of decimals presented across the manuscript.
Generally well written, please see my comments in the previous section.
Reviewer 2 Report
Validating fall risk assessment tools is always a challenge. The authors do a nice job looking at the predictive accuracy of three tools in a specific setting. However, there are multiple challenges in how the research study is presented and there is also a question of the clinical meaning and application of the results.
The introduction needs significant editing to clarify the purpose of the research study. Is it to validate the WH-STRATIFY? Or is it specifically to determine if the cut-points for the falls tools for sub-acute care apply to GEMs. If it is the second, there needs to be more description of what a GEMS unit is and also clarity as to why this study is important considering on line 62 there is the statement that the global consensus is to NOT use a numerical risk score. I agree with the global consensus, when you are working with frail older adults, they are all at risk and efforts should be focused on identifying and managing risk vs. a number. Since you state this in the intro, you need clarification as to why, even though global consensus does not support a numerical number, you did the study looking at numerical numbers. It also needs to be made more clear that 2 of the 3 tools already have been tested.
Aims - Clearly stated but there needs to be a clear justification as to why you selected a GEMS unit and what a GEMs unit is so that the result can be applied.
Results/Analysis - the result are clearly stated and the analysis is appropriate
Discussion: The authors make a fantastic point that the accuracy of the tools depend on the patient population, and the patient populations are variable. This actually could be another paper all together that could be a great contribution to the field. If the tools are dependent on the population, and the population is variable, then are these tools actually valid?
Methods: Line 275 - I could not locate the the falls risk factors and score on table 1 - that info is important.
Conclusion - you need to clarify how a numerical score can guide a targeted intervention
Reviewer 3 Report
The study shows important shortcomings in describing the methodology. I consider this a vitally important aspect, so I do not consider it suitable for publication.

Reviewer 4 Report
The paper compared three fall-prediction methods: PH-FRAT, TNH-STRATIFY, and WH-STRATIFY with a group of patients. They found that TNH-STRATIFY and WH-STRATIFY are much better than PH-FRAT. PH-FRAT failed to detect the fall risks for the patients.
This paper considered these 3 assessments as a whole score. However, these three assessments must have scored the patients based on many factors. If the authors can analyze individual factors in these assessments, the paper will be much better.
The patients with higher scores may take more care to prevent their falling in their daily life. Is it possible that the caution may decrease the falling rate and affect the assessment?
Round 2
Reviewer 1 Report
Authors should be congratulated for addressing the comments.
Author Response
Thank you for your comments
Reviewer 2 Report
Hi - thanks for addressing concerns in the first review. I still feel the challenge with this work is that you are proposing two things - introducing a new assessment tool and looking at the accuracy of the three tools in a clinical setting. Your revisions have improved this but it is still a little complicated to follow. I also think your key finding is "Overall, all tools had low predictive accuracy for falls". which is stated in the last sentence of the results section of your abstract. I would like to see that statement addressed in your discussion. See attached doc for specific recommendations/comments

Author Response
please see the attchment

Reviewer 3 Report
After the changes presented, I consider the work to be of a sufficient level to be published in the journal.
Author Response
Thank you for your comments
Round 3
Reviewer 2 Report
Hi,
Thank you for all of your hard work on this manuscript and the edits you have made. I appreciate the effort!